# Continual learning in Deep Networks: an Analysis of the Last Layer

## Abstract

We study how different output layers in a deep neural network learn and forget in continual learning settings. The following three factors can affect catastrophic forgetting in the output layer: (1) weights modifications, (2) interference, and (3) projection drift. In this paper, our goal is to provide more insights into how changing the output layers may address (1) and (2). Some potential solutions to those issues are proposed and evaluated here in several continual learning scenarios. We show that the best-performing type of the output layer depends on the data distribution drifts and/or the amount of data available. In particular, in some cases where a standard linear layer would fail, it turns out that changing parameterization is sufficient in order to achieve a significantly better performance, whithout introducing a continual-learning algorithm and instead using the standard SGD to train a model. Our analysis and results shed light on the dynamics of the output layer in continual learning scenarios, and suggest a way of selecting the best type of output layer for a given scenario.

## 1 Introduction

Continual deep learning algorithms usually rely on end-to-end training of deep neural networks, making them difficult to analyze given the complexity (non-linearity, non-convexity) of the training dynamics. In this work, we instead propose to evaluate the ability of a network to learn or forget in a simplified learning setting: We decompose our model as a *feature extractor* part consisting of all but the final layer, and a *classifier* part which is given by the output layer. Using this decomposition, we can isolate the role of the output layer parameterization in continual learning scenarios.

Recent works on continual learning (CL) point out some drawbacks of parameterizing the output layer as a linear layer, especially in incremental settings (Wu et al., 2019; Zhao et al., 2020; Hou et al., 2019). Among these problems is the unbalance of bias and/or norm of the output layer's vectors. These works propose some solutions but do not study them independently from the feature extractor. Indeed, in CL scenarios, the function learned by the feature extractor changes, and therefore the embedding space changes accordingly; we call this change the *projection drift*. This drift characterizes the change through time of the embedding on a given observation.

By isolating the output layer from the feature extractor, we can study it in a controlled environment, i.e., without projection drifts. Decoupling the feature extractor from a sub-network has shown to be helpful, e.g., in reinforcement learning (Lesort et al., 2018; Raffin et al., 2019; Stooke et al., 2020).

We now list our contributions:
• We propose an evaluation of a large panel of output layer types in incremental, lifelong, and mixed continual scenarios. We show that depending on the type of scenario, the best output layer may vary.

• We describe the different sources of performance decrease in continual learning for the output layer: forgetting, interference, and projection drifts.

• We propose different solutions to address catastrophic forgetting in the output layer: a simplified weight normalization layer, two masking strategies, and an alternative to Nearest Mean Classifier using median vectors.

## 2  RELATED WORKS

In most deep continual learning papers, the last layer of the deep neural network is implemented as a linear layer as in typical classifier architectures in i.i.d. settings such as VGG (Simonyan & Zisserman, 2014), ResNet (He et al., 2016), and GoogleNet (Szegedy et al., 2014). Recently, several CL approaches questioned this approach, showing that the norm or the bias can be unbalanced for the last classes observed in class-incremental settings. For example, BIC (Wu et al., 2019) proposed to train two additional parameters using a subset of the training dataset after training the rest of the model to correct for the unbalance of the bias. In the same spirit, (Zhao et al., 2020) proposed to compute a *Weight Aligning* value to apply to vector to balance the norm of past classes vectors and the new classes vectors of a linear layer. (Zhao et al., 2020) also experiments with a layer, the weight normalization layer (Salimans & Kingma, 2016), that decouples the norm of the weight matrix from its direction in order to normalize the vector of linear output layers. (Hou et al., 2019; Caccia et al., 2021) proposed to apply a cosine normalization of the layer to avoid the norm and bias unbalance.

Linear Discriminant Analysis (LDA) Classifiers in incremental settings have been used for dozens of years (Aliyari Ghassabeh et al., 2015; Chatterjee & Roychowdhury, 1997; Demir & Ozmehmet, 2005). The streaming version SLDA from (Shaoning Pang et al., 2005) has recently been revisited in deep continual learning in (Hayes & Kanan, 2020). SLDA combines the mean and the covariance of the training data features to classify new observations. A more straightforward approach, as implemented in iCaRL (Rebuffi et al., 2017) is to average the features for each class and apply a nearest neighbor classifier also called Nearest Mean Classifier (NMC).

In this work, we study the capability of those various types of top layers with a fixed pre-trained model in incremental and lifelong settings as defined in (Lesort et al., 2021). Continual learning research field studied many strategies, e.g. rehearsal, generative replay, dynamic architectures or regularization. In this study, we do not aim at modifying the training process or proposing a new one. We study how different layer types learn in various continual scenarios without a CL approach.

## 3  OUTPUT LAYER TYPES

We here review the output layer parameterizations studied in this work. We introduce the notations, and present the main characteristics of linear layers. Then, we explain how standard layer parameterization struggles to learn continually, and we present different parameterizations to overcome the problems. Finally, we present several other types of output layers not trained by gradient descent.

### 3.1  NOTATIONS

We will study functions $f_\theta(\cdot)$ parameterized by a vector of parameters $\theta$ representing the set of weight matrices and bias vectors of a deep network. In continual learning, the goal is to train $f_\theta(\cdot)$ on a sequence of task $[\mathcal{T}_0, \mathcal{T}_1, ..., \mathcal{T}_{T-1}]$, such that $\forall(x_t, y_t) \sim \mathcal{T}_t$ $(t \in [0, T-1])$, $f_\theta(x) = y$.

We can decompose $f_\theta(\cdot)$ into two parts, (1) a feature extractor $\phi_{\theta^-}(\cdot)$ which encodes the observation $x_t$ into a feature vector $z_t$, (2) an output layer $g_{\theta^+}(\cdot)$ which transforms the feature vector into a non-normalized vector $o_t$ (the logits). $\theta^-$ and $\theta^+$ are two complementary subsets of $\theta$.

### 3.2  LINEAR LAYER

A linear layer is parameterized by a weight matrix $A$ and bias vector $b$, respectively of size $N \times h$ and $N$, where $h$ is the size of the latent vector (the activations of the penultimate layer) and $N$ is the number of classes. For $z$ a latent vector, the output layer computes the operation $o = Az + b$. We can formulate this operation for a single class $i$ with $\langle z, A_i \rangle + b_i = o_i$, where $\langle \cdot \rangle$ is the euclidean scalar product, $A_i$ is the $i$th row of the weight matrix viewed as a vector and $b_i$ is the corresponding scalar bias. It can be rewritten:

$$\|z\|\|A_i\| \cdot cos(\angle(z, A_i)) + b_i = o_i \tag{1}$$

Where $\angle(\cdot, \cdot)$ is the angle between two vectors and $\|\cdot\|$ denotes here the euclidean norm of a vector.

Eq. 1 highlights the 3 components that need to be learned to make a correct prediction $\hat{y} = \text{argmax}_i(o_i)$: The norm of $A_i$, its angle with the latent representation, and the bias. In particular,

if the training dynamics makes a particular $\|A_i\|$ or $b_i$ larger than others, then the network will be considerably biased into predicting the class $i$. In incremental learning, the vectors $A_i$ and values $b_i$ are learned sequentially, hence, the model will grow the $\|A_i\|$ and $b_i$ for current classes and reduce them for past classes to ease current task ( Figure 1). This phenomenon may lead to forgetting.

### 3.3 LINEAR CLASSIFIERS IN CONTINUAL LEARNING AND REPARAMETERIZATIONS

In continual learning, assuming that the true function $f(x) = y$ is fixed, there are two main cases of data distribution drift (see e.g. Lesort et al. (2021)): either we get new data of known classes (domain drift $\rightarrow$ lifelong scenario ) or receive new data from new classes (virtual concept drift $\rightarrow$ incremental scenario). For a linear output layer, forgetting is caused by different mechanisms in both scenarios.

**In incremental learning**, the vectors $A_i$ and $b_i$ are learned sequentially (one by one or set by set). When learning a new $A_i$ (and $b_i$), the training algorithm does not have access to past classes, and hence cannot ensure that learning weight vectors for new classes does not interfere with past classes with eq. 1. As an illustration, suppose two classes are alike in the latent space. In that case, their respective latent vectors might be similar, and **interference** might happen between the two classes (see fig. 2), a fortiori if norm and bias are not balanced.

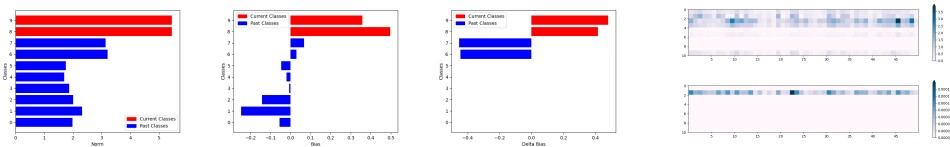

Figure 1: Illustration of norm and bias unbalance at the end of a CIFAR10 continual experiment. 5 tasks with 2 classes each. (left) the norm of each output vector at the end of the last task, (middle left) bias at the end of the last task, and (middle right) the difference between the last task's bias and current bias. We can see a clear unbalance in norm and bias for the last active vectors (classes 8 and 9). The middle right figure shows us that the imbalance of bias is primarily due to the modification of bias from the previous task (classes 6 and 7). (right) Illustration of Forgetting: we plot the difference between the first 50 weights after task 44 and after task 45 in the scenario Core10Mix. Task 45 is composed of one class different from 44's one. Those two figures illustrate the impact of the masking. Indeed it avoids the modification of weights of other classes.

One solution found in the bibliography to avoid interference is to counteract unbalance between the norms of vectors or bias. We can modify eq. 1 in several ways to mitigate such unbalance:

$$\text{Removing the bias (\textit{Linear\_no\_bias} layer):} \quad \rightarrow \quad \|z\|\|A_i\| \cdot cos(\angle(z_t, A_i)) = o_i \tag{2}$$

$$\text{Normalizing output vectors (\textit{WeightNorm} layer):} \quad \rightarrow \quad \|z\| \cdot cos(\angle(z_t, A_i)) = o_i \tag{3}$$

$$\text{Measuring only the angle (\textit{CosLayer}):} \quad \rightarrow \quad cos(\angle(z, A_i)) = o_i \tag{4}$$

WeightNorm (eq. 3) is similar to the original WeightNorm layer (Salimans & Kingma (2016), here denoted by *Original WeightNorm*) experimented in Zhao et al. (2020) in a continual learning context:

$$\gamma_i \|z\| \cdot cos(\angle(z_t, A_i)) + b_i = o_i \tag{5}$$

However, in the original WeightNorm, the additional scaling parameter $\gamma$ and the bias $b$ are learned during training. These parameters are akin to the parameters in BatchNorm layers (Ioffe & Szegedy, 2015), which have been shown to hurt the training of intermediate layers in continual learning (Lomonaco et al., 2020). Hence, our proposed WeightNorm layer (eq. 3) avoids such interference by ensuring a unit norm for all vectors and removing bias and gamma parameters.

**Forgetting might also be caused by weight modification**: this happens when the optimizer modifies weight vectors from past classes that are not present in the current task. In the literature, regularization strategies (e.g. EWC (Kirkpatrick et al., 2017), KFRA (Ritter et al., 2018)) aim at avoiding such forgetting by penalizing modification of important parameters.

We introduce a more radical strategy to avoid that the update on a specific class affects another past one. The strategy consists in masking some classes during the update step. We propose two types of

masking: in *single masking*, we only update weights for the output vector of the true target and in *group masking*, we mask all classes that are not in the mini-batch. With the following strategies, the update step cannot change $A_i$ and $b_i$ of past classes and avoid sub-sequential interference. In Figure 1, we illustrate the impact of *single masking* in a task with only one class from one experiment of the paper. In incremental scenarios, all masking strategies can also be seen as a regularization strategy that strictly forbids the modification of past weights in the last layer.

The masking strategy resembles the update step in a multi-head architecture (van de Ven & Tolias, 2019) for gradient descent since in this case, the gradient is computed only for a subset of selected classes. However, in our setup there is only one head, and the inference is achieved without the help of any external supervision, such as a task label.

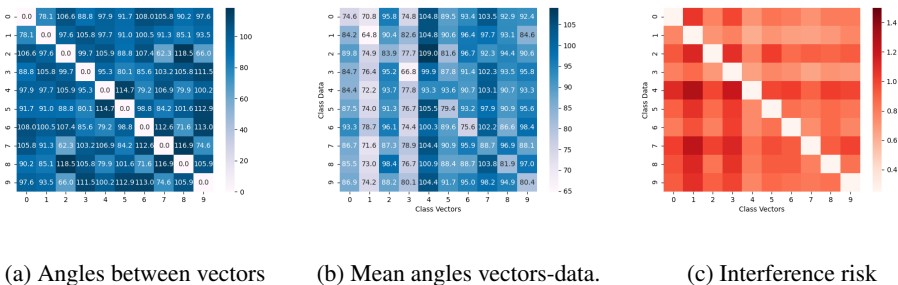

(a) Angles between vectors      (b) Mean angles vectors-data.      (c) Interference risk

Figure 2: Illustration of *interference* for the linear layer: We plot **(left)** the angles (in degree) between the output vectors $A_i$, **(middle)** the mean angle between the latent vectors $z$ of each class and the $A_i$ vectors and **(right)** ratio between the mean angle with wrong classes, and the mean angle of the target class: a high value indicates a higher risk of interference. In this experiment, there is a high risk of interference between examples for class 4 and examples of class 1 or 3, as pictured by a dark red cell. See appendix F for a similar experiment for WeightNorm.

**In lifelong learning**, forgetting happens when new instances of a previously learned class make the output layer forget features that were important for past instances or modify the direction of the output vectors. However, those problems are usually less prone to catastrophic forgetting but may still have a significant impact in complex scenarios. One of the reasons that catastrophic forgetting is less important is that all classes are available simultaneously, which makes it possible for the output vectors to be coordinated and avoid interference.

### 3.4    SIMILARITY-BASED CLASSIFIERS (TRAINED WITHOUT GRADIENT)

To avoid the interference and weight modification, we can use another type of layers that rely on the similarity between training latent vectors and testing latent vectors.

One of the most popular classifiers that can easily be adapted into a continual or online learning setting is the k-nearest neighbors (KNN) classifier. This classifier searches for the k nearest exemplars of a given observation in the training set and predicts the class based on the class of the k neighbors. This classifier's hyper-parameters are the number of neighbors $k$ and the distance used (here the usual euclidean norm).

A lighter strategy is the nearest mean classifier (denoted *MeanLayer*) as proposed in iCaRL (Rebuffi et al., 2017), this strategy consists of saving the mean of the features of each class $k$ while learning (called a *prototype* $\mu_k$) and predict the class given by the nearest mean for test latent vectors.

In this paper, we propose to evaluate the performance of a median classifier (denoted *MedianLayer*), where we replace the mean of the features with the median of the features. The median vector is computed by selecting the median value on each dimension. The idea behind this approach is that the prototype might be more central in the feature distribution and maybe avoid interference with similar classes more effectively.

An alternative is the linear discriminant analysis classifier, which uses both the mean of the features and a covariance. The online version of this algorithm is streaming linear discriminant analysis

(SLDA) (Shaoning Pang et al., 2005). We used the approach with online updates of the covariance matrix of (Dasgupta & Hsu, 2007) as in (Hayes & Kanan, 2020) (see appendix B).

Similarity-based classifiers are very effective in the literature. However, they cannot be directly used to train the feature extractor since they are not differentiable. Hence, they need to store samples from past tasks as rehearsal strategies or be applied in a setting without projection drift.

In this section, we introduced the different output layer types that we will use in experiments. We introduce three modifications: (1) a simplified WeightNorm (referred to as *WeightNorm*), (2) two masking strategies, single masking and group masking, compatible with any reparameterization of the linear layer, and (3) a Nearest Median Classifier.

## 4    EXPERIMENTS

In this section, we will review how the different output layer types, presented in section 3, can solve various continual scenarios. We isolate the output layer from the remaining of the neural network by using a frozen pre-trained model. This methodology consists of creating a scenario where no projection drift occurs. Indeed, projection drifts may only lead to a deterioration of performance. Hence our study can eliminate approaches that would, in all cases, fail under such conditions. Our findings can also be directly transferred to settings where the projection drift is negligible.

In a preliminary experiment, we benchmark all output layer types in an i.i.d. setting to fix some of their hyper-parameters for further continual training. The conclusion of these experiments are in section 4.1, but the full experiments are in appendix section C and in Table 3.

In a second setup, we train the various output layers in continual learning scenarios. Finally, we experiment with a vanilla continual approach close to a rehearsal approach: we train using only on a subset of the training data, and compare layer parameterizations in this low data regime.

### 4.1    DATASETS AND PRE-TRAINED MODELS

We conduct our experiments on CIFAR10/CIFAR100 (Krizhevsky & Hinton, 2009) and Core50 (Lomonaco & Maltoni, 2017). Core50 is a dataset composed of 50 objects from 10 categories (e.g. glasses, phone, cup...), filmed in 11 environments (8 for training and 3 for evaluation). We denote Core10, the 10 classes version where we only predict the category, and Core50, the full 50 classes version. We use both to create various types of scenarios.

The preliminary experiments consist of testing the basic output layer (without masking) to select a learning rate (among 0.1, 0.01 and 0.001 ) and architecture for pre-trained models (among ResNet, VGG16 and GoogleNet available on the torchvision library (Paszke et al., 2019)). While choosing pre-trained models, we made sure that pre-training had not been done using the same dataset that we used in our continual learning experiments (e.g. we did not choose a model pre-trained on CIFAR10 for our CIFAR10 experiments), which would otherwise be "cheating" as the pre-training would have been made using all available data, and not been restricted to data sequentially made available as in continual learning.

Following the results (appendix C), we choose a learning rate of 0.01 for the original linear layer and original linear layer without bias (*Linear_no_bias*) as well as for their masked counterpart and 0.1 for the other layers. Secondly, we found that for Core50 and Core10 experiments, the ResNet model performs best. We also eliminated the CIFAR100 dataset from our incremental experiments because the pre-trained model we tested (pre-trained on CIFAR10) did not permit to achieve a good enough accuracy. We kept CIFAR10, Core10, and Core50 for subsequent experiments. We added CUB200 and a modified version of CIFAR100 (referred to as CIFAR100Lifelong), for continual experiments. Moreover, we see that 5 epochs per task are sufficient on these datasets, and we will keep this number of epochs for all experiments.

### 4.2    CONTINUAL EXPERIMENTS SETTINGS

In these experiments, we evaluate the capacity of every output layer to learn continually with a fixed feature extractor. We evaluate the CL performance on virtual concept drifts (incremental scenario),

on domain drifts (lifelong scenario) (cf section 3.3). Incremental settings consist of a sequence of tasks where all new tasks bring new unknown classes. Lifelong settings consists of a sequence of tasks where each new task brings new examples of known classes. We also evaluate the best layers in a mixed scenario with both kinds of drifts in the end. Our experiments use the single-head framework (Farquhar & Gal, 2018), as opposed to multi-head where the task id is used for inference to pre-select a subset of classes. We do not use task labels during training nor testing. The change in labels can signal a change of tasks. However, it does not trigger any continual mechanisms in our experiments, and all training are done using stochastic gradient descent with a momentum of 0.9.

**Incremental Scenarios: CIFAR10, Core50, CUB200**: Incremental scenarios are characterized by a virtual concept drift between two tasks. These settings evaluate the capacity of learning incremental new classes and distinguishing them from the others. The core50 scenario is composed of 10 tasks; each of them is composed of 5 classes, the CIFAR10 scenario is composed of 5 tasks with 2 classes each, and the CUB200 scenario is composed of 10 tasks of 20 classes each.

**Lifelong Scenarios: Core10Lifelong, CIFAR100Lifelong:**  Lifelong settings are characterized by a domain drift between two tasks. The classes stay the same, but the instances change. These settings evaluate the capacity of improving at classifying with new data.  The **Core10Lifelong** scenario is composed of 8 tasks with 10 classes in a given environment, each new task, we visit a new environment. **CIFAR100Lifelong** is composed of 5 tasks, with 20 labels each that also stay the same. Data are labeled with the coarse labels of CIFAR100. However, data are shared between tasks using the original label to ensure a domain drift between tasks (More detail in appendix in section A).

**Mixed Scenario: Core10Mix:**  In this scenario, a new task is triggered by either a virtual concept drift or a domain drift. In this case, the domain drift is characterized by the transition from one object to another in the same category. This scenario is composed of 50 tasks, where each task corresponds to a new object (with category annotation) in all its training domain. So a new task is always a new object, but it is either data from a new class or from a known class visited through another object.

The results for those experiments are in section 5.1. We report at each epoch the average accuracy over the whole test set of the scenario (with data of all tasks). This measure makes it possible to report the accuracy on a fixed test set and makes the visualization of progress in solving the complete scenario easier. For easy reproducibility, we put in appendix H, how the scenarios are created with the *continuum* library (Douillard & Lesort, 2021).

### 4.3 SUBSET EXPERIMENTS SETTINGS

In many continual learning approaches, a subset of the training is saved, either for rehearsal purposes or for validation purposes (Buzzega et al., 2020; Douillard et al., 2020; Prabhu et al., 2020). In this experiment, we evaluate the capacity of each layer to be trained from scratch with such subsets. It could be more efficient to continually train the feature extractor in a continual end-to-end setting and learn a suitable output layer afterward from a subset of data. This procedure is very fast, and it avoids all problems that projection drifts or any other drift can create. The only (not so simple) problem to solve is which examples to store and how many. Nevertheless, the successes in few-shot learning (Lake et al., 2011; Fei-Fei et al., 2006; Wang et al., 2019) or fast adaptation (Caccia et al., 2020) show that when the feature space is good enough, only a small set of data is enough to identify a class.

## 5 RESULTS

### 5.1 CONTINUAL EXPERIMENTS

We split the results of continual experiments into two groups of figures; the first group (Fig. 3) only compares the various parameterizations of a linear layer presented in section 3. The second group ((Fig. 4)) compares the best performing layers of the first group (CosLayer and WeightNorm) to layers that are not trained by gradient descent, namely: KNN, MeanLayer, MedianLayer, and SLDA. The results presented in fig. 3 and fig. 4 lead us to the following conclusions:

**Among the reparameterizations of linear layers, *Coslayer* and *WeightNorm* are the overall best performing.** Indeed, in incremental settings (figs. 3a, 3c, 3e), they are always among the best performing methods. Both of them only rely on the angle between data and output vectors, which

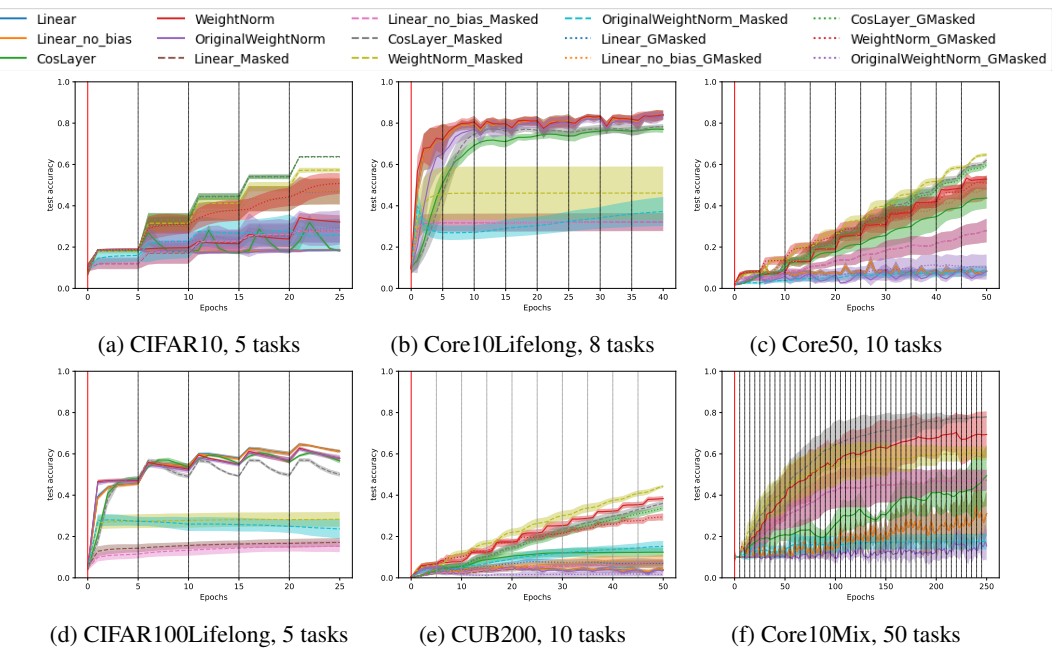

(a) CIFAR10, 5 tasks   (b) Core10Lifelong, 8 tasks   (c) Core50, 10 tasks

(d) CIFAR100Lifelong, 5 tasks   (e) CUB200, 10 tasks   (f) Core10Mix, 50 tasks

Figure 3: Experiments realized on 8 different task orders. We plot the test accuracy on the full test set for each epoch. We compare the different parameterizations of the linear layer. A vertical line represent a task transition. The notation _Masked denotes layer trained with single masking while _GMasked denotes layer trained with group masking

reduces the risk of interference as described in section 3. The only difference between these layers is that in WeightNorm, the predictions are weighted by the norm of the latent vectors (cf Coslayer eq. 4, WeightNorm eq. 3), which scales the gradients thus may change the learning dynamics.

**Masking is efficient in incremental settings to avoid weight modifications.** In incremental scenarios (figs. 3a, 3c, 3e) the best performing methods are all masked. Except in CUB200 (Fig. 3e) where (only) the WeightNorm layer outperforms the masked layers. It shows that in incremental settings, avoiding weight modification with masking can significantly improve results. However, **single masking[1] creates instability in lifelong scenarios** (figs. 3d, 3b). Masking may prevent a good organization of output vectors together and make the training unstable.

**Removing the bias on the linear layer has no effect.** We did not observe any improvement by removing the bias from the linear layers in our experiments (Linear and Linear_no_bias almost coincide). This questions Wu et al. (2019)'s observations that correcting the bias can mitigate catastrophic forgetting. Nevertheless, we hypothesize that the effect of the bias in continual learning depends on the task at hand.

**Similarity-based layers perform well regardless of the data drift type** (fig. 4). These layers are easy to train since they need to see the data only once to achieve their best performance. In fig. 4 these layers look to perform very well in all settings in comparison with linear layers, especially SLDA. Nevertheless, we should not forget that they cannot be used to train an end-to-end neural network. They could, however, be associated with another layer to train the feature extractor as in iCARL (Rebuffi et al., 2017).

**In lifelong, we cannot avoid weight modification with masking.** In lifelong scenarios, tasks use the same output vectors, i.e., the same weights; therefore, freezing weights forbids further learning. In Core10Lifelong setting (fig. 3b), weight modification does not look to be a problem, and no decrease in performance indicates catastrophic forgetting, while in CIFAR100Lifelong (fig. 3d) all layers seem to forget while learning a new task. This phenomenon is due to the similarity of the different modes

---

[1]We removed the layer masked by group in the lifelong experiment because it is useless when all classes are available simultaneously.

Table 1: Subset experiments: Mean Accuracy and standard deviation on 8 runs with different seeds.

| OutLayer | Dataset \ Subset | 100 | 200 | 500 | 1000 | All |
|---|---|---|---|---|---|---|
| CosLayer | Core50 | $10.30_{\pm1.98}$ | $12.43_{\pm3.56}$ | $18.11_{\pm4.17}$ | $26.48_{\pm4.16}$ | $53.93_{\pm0.90}$ |
| WeightNorm | Core50 | $32.06_{\pm2.93}$ | $44.32_{\pm1.92}$ | $61.91_{\pm1.10}$ | $69.44_{\pm0.67}$ | $77.04_{\pm0.33}$ |
| OriginalWeightNorm | Core50 | $33.19_{\pm2.62}$ | $45.68_{\pm1.96}$ | $62.39_{\pm1.01}$ | $69.17_{\pm0.65}$ | $76.85_{\pm0.61}$ |
| CosLayer-Masked | Core50 | $30.62_{\pm2.57}$ | $41.22_{\pm1.68}$ | $56.47_{\pm1.13}$ | $63.10_{\pm0.62}$ | $63.43_{\pm5.78}$ |
| WeightNorm-Masked | Core50 | $12.95_{\pm2.07}$ | $14.98_{\pm2.89}$ | $21.13_{\pm1.22}$ | $23.42_{\pm2.49}$ | $31.18_{\pm17.50}$ |
| OriginalWeightNorm-Masked | Core50 | $12.27_{\pm2.59}$ | $17.42_{\pm2.52}$ | $20.82_{\pm4.94}$ | $22.21_{\pm5.50}$ | $22.43_{\pm4.36}$ |
| Linear | Core50 | $32.80_{\pm2.67}$ | $44.88_{\pm1.94}$ | $61.80_{\pm1.06}$ | $69.03_{\pm0.65}$ | $76.29_{\pm0.34}$ |
| Linear-no-bias | Core50 | $32.80_{\pm2.68}$ | $44.89_{\pm1.94}$ | $61.80_{\pm1.06}$ | $69.02_{\pm0.65}$ | $76.29_{\pm0.34}$ |
| Linear-Masked | Core50 | $7.37_{\pm1.98}$ | $8.20_{\pm2.68}$ | $14.02_{\pm5.88}$ | $18.46_{\pm2.60}$ | $27.25_{\pm28.39}$ |
| Linear-no-bias-Masked | Core50 | $7.36_{\pm1.97}$ | $8.22_{\pm2.67}$ | $14.04_{\pm5.88}$ | $18.48_{\pm2.55}$ | $25.81_{\pm29.13}$ |
| KNN | Core50 | $25.95_{\pm3.09}$ | $34.17_{\pm2.14}$ | $45.50_{\pm2.04}$ | $52.13_{\pm1.19}$ | $65.50_{\pm0.14}$ |
| SLDA | Core50 | $17.30_{\pm1.23}$ | $30.23_{\pm5.58}$ | $17.71_{\pm1.47}$ | $60.14_{\pm0.93}$ | $78.55_{\pm0.03}$ |
| MeanLayer | Core50 | $28.81_{\pm2.68}$ | $40.71_{\pm1.65}$ | $56.52_{\pm1.29}$ | $63.20_{\pm0.79}$ | $71.51_{\pm0.00}$ |
| MedianLayer | Core50 | $26.83_{\pm2.26}$ | $36.03_{\pm1.65}$ | $53.07_{\pm1.15}$ | $60.73_{\pm0.77}$ | $70.22_{\pm0.00}$ |

of the data distribution, in Core10Lifelong only the background changes from one task to another. At the same time, in CIFAR100Lifelong, the data are very different from one task to another, i.e., for the coarse label "aquatic mammals" the data go from beavers to dolphins to otters to seals to finally whales in separate tasks.

**Linear layers do not perform similarly while exposed to different types of drift.** A tendency that emerges from these experiments is that all layers are compatible with lifelong learning even if masking may create some instability. In incremental and mixed scenarios, the layers relying only on angles between data and output vectors and masking helps to avoid weight modification.

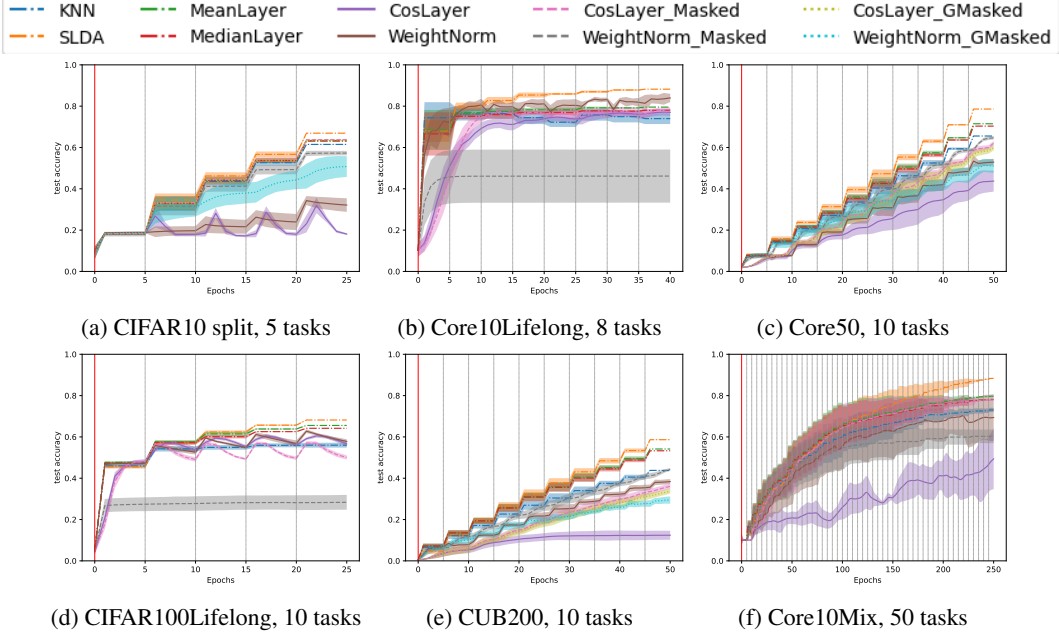

Figure 4: Comparison between layers trained by gradient descent and layers trained without gradients. Experiments realized on 8 different task orders (cf Appendix G). We plot the full test accuracy.

## 5.2 TRAINING WITH A SUBSET

In these experiments, we randomly select a sub-sample of the training dataset, and we train the output layers on it. Experiment's results are gathered in Table 1 (Core50) and Table 4 (other datasets). The goal is to measure how the layers previously experimented can learn with a low amount of data. It gives us also a good continual baseline close to the GDump approach (Prabhu et al., 2020) where we save samples in the memory, and we train at the end of the scenario only on the memory's samples.

This approach performs particularly well in comparison to many continual approaches. Here are some conclusions of these experiments:

**Masking does not work in an iid setting.** In table 1, we can note that most of the masked layers underperform in this setting, except the CosLayer, which performs better with a mask.

**WeightNorm is still very competitive among linear layers.** In most of the experiments, *WeightNorm* is among the best reparameterization of the linear layer.

**Similarity-based layers tend to slightly underperform linear layers.** In most of the results, these layers do not perform as well as in the continual scenarios, even if they can still have good accuracy.

***MedianLayer* does not show much interest in comparison with *MeanLayer*.** MedianLayer works quite well in our experiment but shows no real advantages against MeanLayer; moreover, Mean-Layer is easier to compute online. In a dataset with many outliers, MeanLayer could show some advantages against the MeanLayer. In our experiments, only training on CIFAR10 with all data made MedianLayer better than MeanLayer (cf appendix Table 4).

These experiments with iid training on a low data regime show that in such settings, reparameterization of the linear layer are well-performing. In conclusion of all the experiments, we recommend using WeigthNorm (especially in lifelong) or CosLayer in its masked version in continual scenarios.

## 6 DISCUSSION

In this paper, we studied and proposed various approaches that can help to avoid catastrophic forgetting in the output layer by addressing weights modification and interference, by restricting ourself to a setup where there is no projection drift.

On the opposite, multi-head architectures use the task label at test time to train a head specific to the current task. It avoids interference and weight modification by training separate heads. Interestingly, in multi-heads settings, using regularization (Kirkpatrick et al., 2017; Zenke et al., 2017; Ritter et al., 2018) or dynamic architecture (Rusu et al., 2016; Mallya et al., 2018; Rajasegaran et al., 2019), have been shown to perform well on end-to-end tasks. Then, they are able to deal with projection drifts: the head of task 0 still works when applied to the feature extractor trained up until the last task.

Unfortunately, end-to-end approaches that work in multi-heads can not be directly connected with our findings to create a good end-to-end single head approach. Indeed, single-head models need discriminative inter-task representations that are not necessary for multi-heads settings. Therefore, the constraints on the feature extractor are not the same, and direct transfer is not possible.

Dealing with projection drift in single-head settings stays then one of the biggest challenge in continual learning. Replay might be a simple solution to simulate an iid training and avoid problems with projection drift. However, even if replay might be necessary in continual learning scenarios (Diethe et al., 2018; Lesort et al., 2019), replaying data may have consequences in the memory needs and compute needs. Therefore, it is worth finding training procedures, architectures, or regularizations that do not necessarily replace replay but at least reduce its need.

## 7 CONCLUSION

In this paper, we conduct an empirical evaluation of the various output layers of deep neural networks in CL scenarios. This evaluation gives us clear insights into how output layers learn continually or on a low data regime. We also showed how data distribution drifts might affect the output layers differently, namely in lifelong and incremental settings. This perspective should incite researchers to study more the connection between data distribution drifts characteristics and catastrophic forgetting.

On one hand, classifiers such as KNN, MeanLayer and SLDA provide very strong baselines that are very effective with a frozen feature extractor. On the other hand, our results show that by a reparameterization of the classical linear layer, we can greatly improve performance, especially in incremental settings. This reparameterization helps to reduce the risks of interference and weight modifications. Some output layer types can learn continually when there is no projection drift with stochastic gradient training. These findings could be integrated in end-to-end training to make continual training more efficient.

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

## APPENDIX

## A   SCENARIOS DESCRIPTION

### A.1   CIFAR10 5 TASKS

A classical incremental setting, with 2 classes per task with original CIFAR10 dataset.

#### A.1.1   CIFAR100LIFELONG 5 TASKS

CIFAR100 dataset has two labelization systems: the classes labels and the coarse labels. There are 20 coarse labels gathering 5 classes each. To create CIFAR100Lifelong 5 tasks, we share data into tasks such as having one data for each coarse label in each task. On the other hand, we use class labels to ensure that the data is different for each task.

We have then 5 tasks with data for all coarse labels and from 20 different classes. A class does not appear in 2 tasks, as in an incremental. However, the model should predict the coarse label. We obtain a scenario with 5 tasks with always the same 20 labels. However, the data change, and the model should learn continually to predict the coarse labels for data for all classes correctly. This scenario makes it possible to measure the resistance to mode change (change of class) while continually learning the coarse label classification.

### A.2   CORE50 10 TASKS

A classical incremental setting, with 5 classes per task with the 50 objects core50.

### A.3   CORE10LIFELONG 8 TASKS

Core50 is composed of 50 objects, equitably shared into 10 classes filmed in 11 environments (8 environments for train and 3 for tests.) In Core10Lifelong, we have 8 tasks where we explore each time one environment with all objects. Objects are annotated with their class labels. Hence, we have 8 tasks of 10-way classification. The object stays the same from one task to another; only the environment changes. This scenario makes it possible to measure the resistance to background change while learning continually.

### A.4   CORE10MIX 50 TASKS

In this scenario, each task contains the data corresponding to one object in all of its environments. We have hence a sequence of 50 tasks. However, objects are annotated with their classes labels. Hence all classes are revisited 5 times in the scenario (in a random way). This scenario is a mixture of incremental and lifelong to measure the ability to solve a scenario with several types of data distribution drifts.

## B   SLDA LAYER

The approach we use (proposed by Hayes & Kanan (2020)) stores one mean vector per class $\mu_k$ with a counter $c_k \in \mathbb{R}$ as well as a covariance matrix of the same size as $A$ (common to all classes). The covariance matrix's update is done with:

$$\Sigma_{t+1} = \frac{t\Sigma_t + \Delta_t}{t + 1} \tag{6}$$

Where $\Delta_t$ is:

$$\Delta_t = \frac{t(z_t - \mu_k)^2}{t + 1} \tag{7}$$

For prediction, SLDA uses a precision matrix $\Lambda = [(1 - \epsilon)\Sigma + \epsilon I]^{-1}$, where $\epsilon = 10^{-4}$ is called the shrinkage parameter and $I \in \mathbb{R}^{d \times d}$ is the identity matrix.

Then two vectors $w$ and $b$ are computed $w_k = \Lambda \mu_k$, $b_k = -\frac{1}{2}(\mu_k^\top \Lambda \mu_k)$

And finally: $o_k = \langle z_t, w_k \rangle + b_k$

## C Preliminary Experiments

### C.1 Preliminary Experiments Setting

The preliminary experiments are designed to see how the different output layers work without continual learning constraints. This experiment is also conducted to select the learning rate and architecture for further experiments.

We experimented with CIFAR10 and CIFAR100 datasets, and we use resnet20 pre-trained model from here [2] (BSD 3-Clause License). The main idea is to compare those results with our results. We take the frozen pre-trained model and replace the output layer with a new one we train on all data. (1) We experiment with training on CIFAR10 with a neural network pre-trained on CIFAR100 and the opposite. (2) We also trained on CIFAR10 with the model pre-trained on CIFAR10 to see if we would recover the original accuracy, and we did the same for CIFAR100. Doing continual learning on a dataset with a model pre-trained on this dataset is not recommended since it means having access to the whole data from the beginning of training. We did not use (2) setting in the further experiments, and it was just used as a reference.

(3) We also experiment also with Core50 dataset (Lomonaco & Maltoni, 2017). We used models pre-trained on ImageNet available on the torchvision library (Paszke et al., 2019). We used VGG16, GoogleNet and Resnet, which have a latent space of dimension 2048, 1024, and 512. We experimented with the Core50 dataset, Core50 setting with 50 classes corresponding to the object id, and Core10 setting with 10 classes corresponding to the object category.

In these preliminary experiments, we evaluate all layers without their masking variant to verify that the pre-trained models were suited to the continual learning scenarios.

### C.2 Preliminary Experiments

The summary of the preliminary results is reported in Table 2. The full results can be found in Table 3. As described in section C.1, we experiment with the simplest output layer for this preliminary experiment, i.e., linear layer, linear layer without bias (Linear_NB), CosLayer, WeightNorm layer, Original WeightNorm (OWeightNorm), KNN, SLDA, and MeanLayer.

The results of those preliminary experiments are that globally the learning rate of 0.1 is adapted to all settings. Secondly, we found that for Core50 and Core10 experiments, the ResNet model was the best suited. Moreover, we see that 5 epochs per task are sufficient to learn a correct solution. Concerning CIFAR experiments, the reported top-1 accuracy on CIFAR10 and CIFAR100 of the original pre-trained models are respectively 92.60%, 69.83%. The settings CIFAR10 pre-trained on CIFAR10 (same for CIFAR100) show us that the training procedure can recover almost the best performance. In CIFAR100, there is a drop of 6% of accuracy, but it stays reasonable in a 100-way classification task.

Even if not very surprising, one interesting result is that training on CIFAR100 with a model pre-trained on CIFAR10 does not work at all. It illustrates that using a pre-trained model is not always a solution in machine learning or continual learning. They should be selected carefully, even if the data might look similar to CIFAR10 and CIFAR100. On the other hand, the experiment on CIFAR10 with CIFAR100 models shows a significant decrease in the performance for all layers.

---

[2] https://github.com/chenyaofo/pytorch-cifar-models

Table 2: Preliminary experiments to select learning rate and architectures with existing type of layers. Experiment realized on 8 seeds $[0, 1, 2, 3, 4, 5, 6, 7]$.

| Dataset | LR | Architecture | Pretraining | Linear | Linear_NB | CosLayer | WeightNorm | OWeightNorm | KNN | SLDA | MeanLayer |
|---|---|---|---|---|---|---|---|---|---|---|---|
| CIFAR10 | 0.1 | resnet | CIFAR10 | $92.55_{\pm0.02}$ | $92.55_{\pm0.02}$ | $92.49_{\pm0.05}$ | $92.56_{\pm0.03}$ | $92.55_{\pm0.04}$ | $91.79_{\pm0.26}$ | $92.37_{\pm0.00}$ | $92.44_{\pm0.00}$ |
| CIFAR10 | 0.1 | resnet | CIFAR100 | $65.37_{\pm2.74}$ | $68.58_{\pm9.55}$ | $66.55_{\pm9.83}$ | $68.55_{\pm0.21}$ | $69.21_{\pm0.36}$ | $60.26_{\pm0.28}$ | $66.90_{\pm0.02}$ | $63.27_{\pm0.00}$ |
| CIFAR100 | 0.1 | resnet | CIFAR10 | $17.32_{\pm0.23}$ | $17.14_{\pm0.22}$ | $13.09_{\pm0.22}$ | $15.11_{\pm0.22}$ | $16.88_{\pm0.30}$ | $14.20_{\pm0.17}$ | $26.44_{\pm0.03}$ | $14.48_{\pm0.00}$ |
| CIFAR100 | 0.1 | resnet | CIFAR100 | $63.97_{\pm0.10}$ | $63.94_{\pm0.11}$ | $60.91_{\pm0.24}$ | $63.62_{\pm0.09}$ | $63.71_{\pm0.15}$ | $60.69_{\pm0.12}$ | $61.56_{\pm0.01}$ | $61.84_{\pm0.00}$ |
| Core50 | 0.1 | resnet | ImageNet | $76.58_{\pm0.41}$ | $76.57_{\pm0.41}$ | $53.93_{\pm0.90}$ | $77.04_{\pm0.33}$ | $76.85_{\pm0.61}$ | $65.50_{\pm0.14}$ | $78.55_{\pm0.03}$ | $71.51_{\pm0.00}$ |
| Core50 | 0.1 | vgg | ImageNet | $71.69_{\pm0.50}$ | $71.44_{\pm0.30}$ | $61.13_{\pm0.39}$ | $72.13_{\pm0.45}$ | $71.53_{\pm0.23}$ | $56.86_{\pm0.08}$ | $69.98_{\pm0.03}$ | $66.67_{\pm0.00}$ |
| Core50 | 0.1 | googlenet | ImageNet | $70.13_{\pm0.27}$ | $70.13_{\pm0.27}$ | $47.81_{\pm3.26}$ | $66.37_{\pm0.74}$ | $69.27_{\pm0.35}$ | $59.45_{\pm0.11}$ | $70.60_{\pm0.11}$ | $60.85_{\pm0.15}$ |
| Core10Lifelong | 0.1 | resnet | ImageNet | $85.45_{\pm0.51}$ | $85.62_{\pm0.58}$ | $78.76_{\pm0.36}$ | $86.29_{\pm0.29}$ | $86.07_{\pm0.51}$ | $75.71_{\pm1.50}$ | $88.06_{\pm0.01}$ | $79.61_{\pm0.00}$ |
| Core10Lifelong | 0.1 | vgg | ImageNet | $85.52_{\pm0.57}$ | $85.34_{\pm0.57}$ | $79.39_{\pm0.18}$ | $85.95_{\pm0.61}$ | $85.56_{\pm0.83}$ | $75.14_{\pm0.88}$ | $83.69_{\pm0.03}$ | $78.55_{\pm0.00}$ |
| Core10Lifelong | 0.1 | googlenet | ImageNet | $85.08_{\pm0.47}$ | $85.08_{\pm0.47}$ | $78.23_{\pm0.33}$ | $83.05_{\pm0.38}$ | $84.97_{\pm0.29}$ | $77.57_{\pm0.96}$ | $86.33_{\pm0.18}$ | $76.92_{\pm0.10}$ |

Table 3: Preliminary experiments to select learning rate and architectures with existing type of layers

| Dataset | LR | Architecture | Pretraining | Linear | Linear_NB | CosLayer | WeightNorm | OWeightNorm | KNN | SLDA | MeanLayer |
|---|---|---|---|---|---|---|---|---|---|---|---|
| CIFAR10 | 0.1 | resnet | CIFAR10 | $92.55_{\pm0.02}$ | $92.55_{\pm0.02}$ | $92.49_{\pm0.05}$ | $92.56_{\pm0.03}$ | $92.55_{\pm0.04}$ | - | - | - |
| CIFAR10 | 0.01 | resnet | CIFAR10 | $92.59_{\pm0.04}$ | $92.59_{\pm0.03}$ | $92.48_{\pm0.03}$ | $92.54_{\pm0.02}$ | $92.60_{\pm0.03}$ | - | - | - |
| CIFAR10 | 0.001 | resnet | CIFAR10 | $92.49_{\pm0.04}$ | $92.49_{\pm0.04}$ | $67.03_{\pm10.29}$ | $92.39_{\pm0.10}$ | $92.57_{\pm0.04}$ | - | - | - |
| CIFAR10 | n/a | resnet | CIFAR10 | - | - | - | - | - | $91.79_{\pm0.26}$ | $92.37_{\pm0.00}$ | $92.44_{\pm0.00}$ |
| CIFAR10 | 0.1 | resnet | CIFAR100 | $65.37_{\pm2.74}$ | $68.58_{\pm9.55}$ | $66.55_{\pm9.83}$ | $68.55_{\pm0.21}$ | $69.21_{\pm0.36}$ | - | - | - |
| CIFAR10 | 0.01 | resnet | CIFAR100 | $75.13_{\pm10.09}$ | $75.13_{\pm10.08}$ | $66.46_{\pm15.03}$ | $68.43_{\pm0.08}$ | $69.27_{\pm0.25}$ | - | - | - |
| CIFAR10 | 0.001 | resnet | CIFAR100 | $73.28_{\pm11.11}$ | $73.30_{\pm11.10}$ | $18.85_{\pm12.54}$ | $60.71_{\pm0.41}$ | $67.79_{\pm0.14}$ | - | - | - |
| CIFAR10 | n/a | resnet | CIFAR100 | - | - | - | - | - | $60.26_{\pm0.28}$ | $66.90_{\pm0.02}$ | $63.27_{\pm0.00}$ |
| CIFAR100 | 0.1 | resnet | CIFAR10 | $17.32_{\pm0.23}$ | $17.14_{\pm0.22}$ | $13.09_{\pm0.22}$ | $15.11_{\pm0.22}$ | $16.88_{\pm0.30}$ | - | - | - |
| CIFAR100 | 0.01 | resnet | CIFAR10 | $14.86_{\pm0.20}$ | $14.86_{\pm0.18}$ | $3.02_{\pm0.43}$ | $12.18_{\pm0.28}$ | $15.20_{\pm0.23}$ | - | - | - |
| CIFAR100 | 0.001 | resnet | CIFAR10 | $6.71_{\pm0.48}$ | $6.71_{\pm0.48}$ | $1.22_{\pm0.33}$ | $2.40_{\pm0.37}$ | $6.77_{\pm0.44}$ | - | - | - |
| CIFAR100 | n/a | resnet | CIFAR10 | - | - | - | - | - | $14.20_{\pm0.17}$ | $26.44_{\pm0.03}$ | $14.48_{\pm0.00}$ |
| CIFAR100 | 0.1 | resnet | CIFAR100 | $63.97_{\pm0.10}$ | $63.94_{\pm0.11}$ | $60.91_{\pm0.24}$ | $63.62_{\pm0.09}$ | $63.71_{\pm0.15}$ | - | - | - |
| CIFAR100 | 0.01 | resnet | CIFAR100 | $63.70_{\pm0.18}$ | $63.71_{\pm0.19}$ | $6.95_{\pm1.19}$ | $62.45_{\pm0.26}$ | $63.96_{\pm0.11}$ | - | - | - |
| CIFAR100 | 0.001 | resnet | CIFAR100 | $54.58_{\pm0.32}$ | $54.54_{\pm0.32}$ | $1.29_{\pm0.25}$ | $21.35_{\pm1.20}$ | $55.54_{\pm0.48}$ | - | - | - |
| CIFAR100 | n/a | resnet | CIFAR100 | - | - | - | - | - | $60.69_{\pm0.12}$ | $61.56_{\pm0.01}$ | $61.84_{\pm0.00}$ |
| Core50 | 0.1 | resnet | ImageNet | $76.58_{\pm0.41}$ | $76.57_{\pm0.41}$ | $53.93_{\pm0.90}$ | $77.04_{\pm0.33}$ | $76.85_{\pm0.61}$ | - | - | - |
| Core50 | 0.01 | resnet | ImageNet | $76.29_{\pm0.34}$ | $76.29_{\pm0.34}$ | $5.80_{\pm0.89}$ | $73.01_{\pm0.51}$ | $76.98_{\pm0.29}$ | - | - | - |
| Core50 | 0.001 | resnet | ImageNet | $62.64_{\pm0.72}$ | $62.65_{\pm0.72}$ | $2.21_{\pm0.56}$ | $30.55_{\pm1.05}$ | $63.42_{\pm0.66}$ | - | - | - |
| Core50 | n/a | resnet | ImageNet | - | - | - | - | - | $65.50_{\pm0.14}$ | $78.55_{\pm0.03}$ | $71.51_{\pm0.00}$ |
| Core50 | 0.1 | vgg | ImageNet | $71.69_{\pm0.50}$ | $71.44_{\pm0.30}$ | $61.13_{\pm0.39}$ | $72.13_{\pm0.45}$ | $71.53_{\pm0.23}$ | - | - | - |
| Core50 | 0.01 | vgg | ImageNet | $72.29_{\pm0.28}$ | $72.29_{\pm0.28}$ | $27.15_{\pm1.66}$ | $71.43_{\pm0.35}$ | $71.91_{\pm0.37}$ | - | - | - |
| Core50 | 0.001 | vgg | ImageNet | $68.79_{\pm0.32}$ | $68.78_{\pm0.31}$ | $3.47_{\pm0.79}$ | $59.54_{\pm0.43}$ | $69.06_{\pm0.32}$ | - | - | - |
| Core50 | n/a | vgg | ImageNet | - | - | - | - | - | $56.86_{\pm0.08}$ | $69.98_{\pm0.03}$ | $66.67_{\pm0.00}$ |
| Core50 | 0.1 | googlenet | ImageNet | $70.13_{\pm0.27}$ | $70.13_{\pm0.27}$ | $47.81_{\pm3.26}$ | $66.37_{\pm0.74}$ | $69.27_{\pm0.35}$ | - | - | - |
| Core50 | 0.01 | googlenet | ImageNet | $65.99_{\pm0.50}$ | $65.99_{\pm0.50}$ | $8.71_{\pm1.31}$ | $61.77_{\pm0.21}$ | $66.65_{\pm0.52}$ | - | - | - |
| Core50 | 0.001 | googlenet | ImageNet | $49.61_{\pm0.85}$ | $49.62_{\pm0.85}$ | $2.55_{\pm0.64}$ | $21.91_{\pm1.17}$ | $49.76_{\pm0.66}$ | - | - | - |
| Core50 | n/a | googlenet | ImageNet | - | - | - | - | - | $59.45_{\pm0.11}$ | $70.60_{\pm0.11}$ | $60.85_{\pm0.15}$ |
| Core10Lifelong | 0.1 | resnet | ImageNet | $85.45_{\pm0.51}$ | $85.62_{\pm0.58}$ | $78.76_{\pm0.36}$ | $86.29_{\pm0.29}$ | $86.07_{\pm0.51}$ | - | - | - |
| Core10Lifelong | 0.01 | resnet | ImageNet | $86.19_{\pm0.33}$ | $86.19_{\pm0.33}$ | $55.26_{\pm1.84}$ | $85.46_{\pm0.40}$ | $86.44_{\pm0.36}$ | - | - | - |
| Core10Lifelong | 0.001 | resnet | ImageNet | $82.53_{\pm0.52}$ | $82.53_{\pm0.52}$ | $13.38_{\pm3.59}$ | $76.42_{\pm1.03}$ | $83.41_{\pm0.45}$ | - | - | - |
| Core10Lifelong | n/a | resnet | ImageNet | - | - | - | - | - | $75.71_{\pm1.50}$ | $88.06_{\pm0.01}$ | $79.61_{\pm0.00}$ |
| Core10Lifelong | 0.1 | vgg | ImageNet | $85.52_{\pm0.57}$ | $85.34_{\pm0.57}$ | $79.39_{\pm0.18}$ | $85.95_{\pm0.61}$ | $85.56_{\pm0.83}$ | - | - | - |
| Core10Lifelong | 0.01 | vgg | ImageNet | $85.85_{\pm0.30}$ | $85.89_{\pm0.27}$ | $72.44_{\pm1.92}$ | $85.37_{\pm0.24}$ | $85.74_{\pm0.65}$ | - | - | - |
| Core10Lifelong | 0.001 | vgg | ImageNet | $84.20_{\pm0.14}$ | $84.20_{\pm0.14}$ | $24.59_{\pm3.17}$ | $82.14_{\pm0.47}$ | $84.51_{\pm0.11}$ | - | - | - |
| Core10Lifelong | n/a | vgg | ImageNet | - | - | - | - | - | $75.14_{\pm0.88}$ | $83.69_{\pm0.03}$ | $78.55_{\pm0.00}$ |
| Core10Lifelong | 0.1 | googlenet | ImageNet | $85.08_{\pm0.47}$ | $85.08_{\pm0.47}$ | $78.23_{\pm0.33}$ | $83.05_{\pm0.38}$ | $84.97_{\pm0.29}$ | - | - | - |
| Core10Lifelong | 0.01 | googlenet | ImageNet | $83.67_{\pm0.26}$ | $83.67_{\pm0.25}$ | $64.27_{\pm2.53}$ | $82.31_{\pm0.28}$ | $83.97_{\pm0.33}$ | - | - | - |
| Core10Lifelong | 0.001 | googlenet | ImageNet | $79.51_{\pm0.36}$ | $79.52_{\pm0.35}$ | $14.85_{\pm2.09}$ | $73.90_{\pm0.63}$ | $80.13_{\pm0.30}$ | - | - | - |
| Core10Lifelong | n/a | googlenet | ImageNet | - | - | - | - | - | $77.57_{\pm0.96}$ | $86.33_{\pm0.18}$ | $76.92_{\pm0.10}$ |

These preliminary experiments make us able to fix some hyper-parameters to study all layers in a common setting. It also offers us interesting insight into the drop of performance in simple experiments when using a pre-trained model instead of training end-to-end. The results of this experiment can be used as a baseline for continual experiments and subsets experiments.

# D  FORGETTING IN LIFELONG SCENARIOS

## D.1  CIFAR100LIFELONG

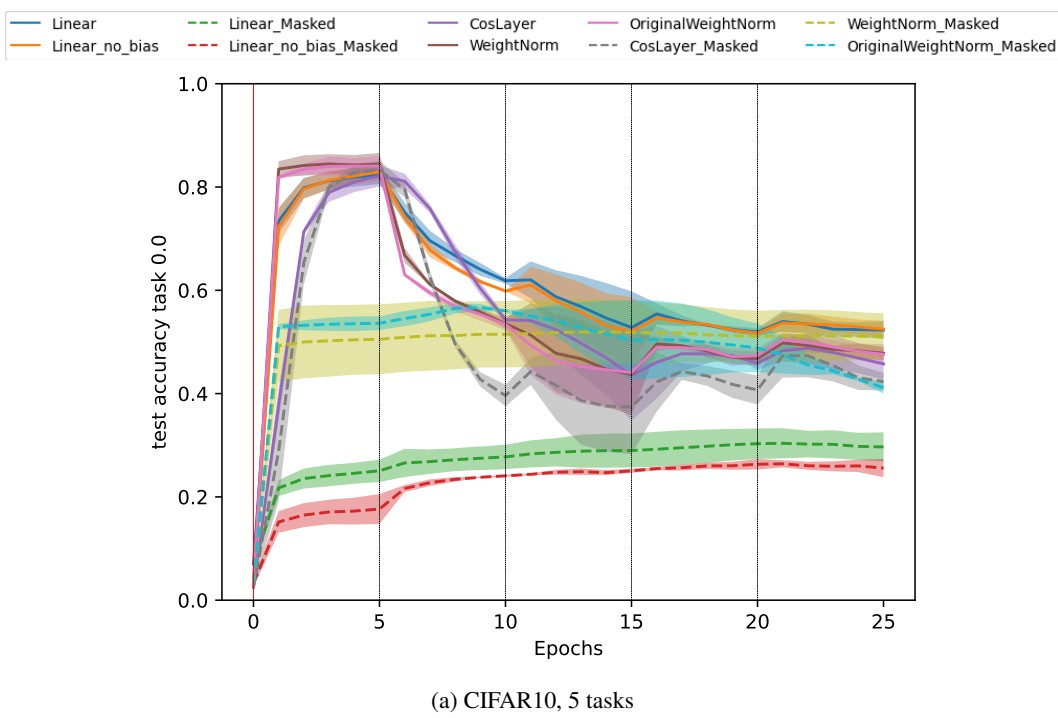

(a) CIFAR10, 5 tasks

Figure 5: Test accuracy on task 0 in CIFAR100Lifelong Scenario.

In Fig. 5, we can see that the model is forgetting the first tasks and the masking have no effect on it. Moreover, since the single masking makes learning less efficient with many classes, it can make the performance bad from the beginning.

## D.2  CORE10LIFELONG

Unfortunately, the Core50 test set does not make it possible to evaluate the forgetting in each task of lifelong scenarios. Indeed, the environments of the test set are different from the train environments. Then it is not possible to evaluate with the test set the evolution of accuracy on past tasks. In the lifelong scenario, we can only evaluate the performance on 3 other unknown environments, similarly to an out of distribution evaluation.

# E    FULL SUBSETS RESULTS

Table 4: Subset experiments: Mean Accuracy and standard deviation on 8 runs with different seeds.

| OutLayer | Dataset \ Subset | 100 | 200 | 500 | 1000 | All |
|---|---|---|---|---|---|---|
| CosLayer | CIFAR10 | $31.81_{\pm5.84}$ | $39.00_{\pm4.90}$ | $48.23_{\pm3.51}$ | $54.09_{\pm2.66}$ | $66.55_{\pm9.83}$ |
| WeightNorm | CIFAR10 | $50.82_{\pm1.91}$ | $56.90_{\pm0.82}$ | $62.64_{\pm0.78}$ | $65.41_{\pm0.49}$ | $68.55_{\pm0.21}$ |
| OriginalWeightNorm | CIFAR10 | $48.22_{\pm2.19}$ | $52.33_{\pm1.53}$ | $55.91_{\pm0.77}$ | $61.66_{\pm0.74}$ | $69.21_{\pm0.36}$ |
| CosLayer-Masked | CIFAR10 | $53.45_{\pm1.63}$ | $58.54_{\pm0.54}$ | $61.25_{\pm0.67}$ | $62.51_{\pm0.46}$ | $67.36_{\pm9.52}$ |
| WeightNorm-Masked | CIFAR10 | $36.55_{\pm3.26}$ | $39.04_{\pm2.23}$ | $41.23_{\pm2.15}$ | $42.44_{\pm1.62}$ | $33.89_{\pm5.33}$ |
| OriginalWeightNorm-Masked | CIFAR10 | $22.79_{\pm10.50}$ | $22.68_{\pm9.30}$ | $21.07_{\pm9.97}$ | $22.45_{\pm6.73}$ | $33.18_{\pm3.31}$ |
| Linear | CIFAR10 | $49.13_{\pm2.31}$ | $54.31_{\pm1.43}$ | $60.16_{\pm0.72}$ | $64.02_{\pm0.77}$ | $75.13_{\pm10.09}$ |
| Linear-no-bias | CIFAR10 | $49.11_{\pm2.27}$ | $54.31_{\pm1.42}$ | $60.15_{\pm0.73}$ | $64.04_{\pm0.77}$ | $75.13_{\pm10.08}$ |
| Linear-Masked | CIFAR10 | $23.02_{\pm8.05}$ | $25.78_{\pm5.89}$ | $27.45_{\pm3.77}$ | $30.02_{\pm4.71}$ | $42.57_{\pm29.59}$ |
| Linear-no-bias-Masked | CIFAR10 | $23.01_{\pm8.12}$ | $25.96_{\pm6.00}$ | $27.74_{\pm3.64}$ | $30.01_{\pm4.84}$ | $39.20_{\pm30.87}$ |
| KNN | CIFAR10 | $45.21_{\pm2.03}$ | $49.47_{\pm1.47}$ | $54.19_{\pm1.13}$ | $57.00_{\pm0.58}$ | $60.26_{\pm0.28}$ |
| SLDA | CIFAR10 | $49.23_{\pm22.67}$ | $61.51_{\pm16.98}$ | $68.47_{\pm13.43}$ | $70.47_{\pm12.44}$ | $66.90_{\pm0.02}$ |
| MeanLayer | CIFAR10 | $62.34_{\pm17.28}$ | $66.58_{\pm14.84}$ | $68.65_{\pm13.71}$ | $69.58_{\pm13.23}$ | $63.27_{\pm0.00}$ |
| MedianLayer | CIFAR10 | $60.65_{\pm18.27}$ | $65.50_{\pm15.53}$ | $63.59_{\pm10.88}$ | $68.84_{\pm13.61}$ | $70.44_{\pm12.72}$ |
| CosLayer | CIFAR100Lifelong | $28.42_{\pm4.38}$ | $36.61_{\pm5.10}$ | $47.39_{\pm3.01}$ | $53.30_{\pm1.87}$ | $65.42_{\pm0.28}$ |
| WeightNorm | CIFAR100Lifelong | $49.19_{\pm2.30}$ | $57.33_{\pm1.14}$ | $63.67_{\pm0.65}$ | $66.39_{\pm0.44}$ | $70.52_{\pm0.22}$ |
| OriginalWeightNorm | CIFAR100Lifelong | $47.68_{\pm2.52}$ | $55.16_{\pm1.10}$ | $60.70_{\pm0.61}$ | $62.83_{\pm0.69}$ | $70.53_{\pm0.19}$ |
| CosLayer-Masked | CIFAR100Lifelong | $53.27_{\pm1.74}$ | $58.88_{\pm0.76}$ | $62.88_{\pm0.54}$ | $64.56_{\pm0.33}$ | $65.74_{\pm0.06}$ |
| WeightNorm-Masked | CIFAR100Lifelong | $23.17_{\pm3.88}$ | $26.00_{\pm5.85}$ | $27.69_{\pm5.08}$ | $28.60_{\pm6.01}$ | $30.78_{\pm4.98}$ |
| OriginalWeightNorm-Masked | CIFAR100Lifelong | $16.13_{\pm8.01}$ | $18.49_{\pm8.51}$ | $16.16_{\pm7.56}$ | $18.12_{\pm6.86}$ | $24.80_{\pm5.16}$ |
| Linear | CIFAR100Lifelong | $47.84_{\pm2.81}$ | $55.88_{\pm1.04}$ | $62.46_{\pm0.79}$ | $65.56_{\pm0.50}$ | $70.36_{\pm0.10}$ |
| Linear-no-bias | CIFAR100Lifelong | $47.24_{\pm2.64}$ | $54.31_{\pm1.42}$ | $62.35_{\pm0.70}$ | $65.38_{\pm0.48}$ | $70.28_{\pm0.29}$ |
| Linear-Masked | CIFAR100Lifelong | $11.41_{\pm4.22}$ | $14.29_{\pm3.81}$ | $14.90_{\pm2.85}$ | $14.76_{\pm2.52}$ | $16.39_{\pm2.11}$ |
| Linear-no-bias-Masked | CIFAR100Lifelong | $11.81_{\pm3.97}$ | $12.41_{\pm3.47}$ | $12.77_{\pm2.43}$ | $13.16_{\pm2.30}$ | $14.53_{\pm2.80}$ |
| KNN | CIFAR100Lifelong | $44.78_{\pm2.39}$ | $52.98_{\pm1.33}$ | $61.87_{\pm0.85}$ | $65.93_{\pm0.77}$ | $66.71_{\pm1.01}$ |
| SLDA | CIFAR100Lifelong | $37.03_{\pm2.04}$ | $55.02_{\pm0.85}$ | $63.07_{\pm0.44}$ | $65.75_{\pm0.28}$ | $68.17_{\pm0.01}$ |
| MeanLayer | CIFAR100Lifelong | $51.63_{\pm2.17}$ | $58.01_{\pm1.05}$ | $62.32_{\pm0.45}$ | $64.13_{\pm0.32}$ | $65.55_{\pm0.00}$ |
| MedianLayer | CIFAR100Lifelong | $48.23_{\pm1.74}$ | $55.30_{\pm1.46}$ | $60.44_{\pm0.35}$ | $62.76_{\pm0.37}$ | $64.17_{\pm0.00}$ |
| CosLayer | Core50 | $10.30_{\pm1.98}$ | $12.43_{\pm3.56}$ | $18.11_{\pm4.17}$ | $26.48_{\pm4.16}$ | $53.93_{\pm0.90}$ |
| WeightNorm | Core50 | $32.06_{\pm2.93}$ | $44.32_{\pm1.92}$ | $61.91_{\pm1.10}$ | $69.44_{\pm0.67}$ | $77.04_{\pm0.33}$ |
| OriginalWeightNorm | Core50 | $33.19_{\pm2.62}$ | $45.68_{\pm1.96}$ | $62.39_{\pm1.01}$ | $69.17_{\pm0.65}$ | $76.85_{\pm0.61}$ |
| CosLayer-Masked | Core50 | $30.62_{\pm2.27}$ | $41.22_{\pm1.68}$ | $56.47_{\pm1.13}$ | $63.10_{\pm0.62}$ | $63.43_{\pm5.78}$ |
| WeightNorm-Masked | Core50 | $12.95_{\pm2.07}$ | $14.98_{\pm2.89}$ | $21.13_{\pm1.22}$ | $23.42_{\pm2.49}$ | $31.18_{\pm17.50}$ |
| OriginalWeightNorm-Masked | Core50 | $12.27_{\pm2.59}$ | $17.42_{\pm2.52}$ | $20.82_{\pm4.94}$ | $22.21_{\pm5.50}$ | $22.43_{\pm4.36}$ |
| Linear | Core50 | $32.80_{\pm2.67}$ | $44.88_{\pm1.94}$ | $61.80_{\pm1.06}$ | $69.03_{\pm0.65}$ | $76.29_{\pm0.34}$ |
| Linear-no-bias | Core50 | $32.80_{\pm2.68}$ | $44.89_{\pm1.94}$ | $61.80_{\pm1.06}$ | $69.02_{\pm0.65}$ | $76.29_{\pm0.34}$ |
| Linear-Masked | Core50 | $7.37_{\pm1.98}$ | $8.20_{\pm2.68}$ | $14.02_{\pm5.88}$ | $18.46_{\pm2.60}$ | $27.25_{\pm28.39}$ |
| Linear-no-bias-Masked | Core50 | $7.36_{\pm1.97}$ | $8.22_{\pm2.67}$ | $14.04_{\pm5.88}$ | $18.48_{\pm2.55}$ | $25.81_{\pm29.13}$ |
| KNN | Core50 | $25.95_{\pm3.09}$ | $34.17_{\pm2.14}$ | $45.50_{\pm2.04}$ | $52.13_{\pm1.19}$ | $65.50_{\pm0.14}$ |
| SLDA | Core50 | $17.30_{\pm1.23}$ | $30.23_{\pm5.58}$ | $17.71_{\pm1.47}$ | $60.14_{\pm0.93}$ | $78.55_{\pm0.03}$ |
| MeanLayer | Core50 | $28.81_{\pm2.68}$ | $40.71_{\pm1.65}$ | $56.52_{\pm1.29}$ | $63.20_{\pm0.79}$ | $71.51_{\pm0.00}$ |
| MedianLayer | Core50 | $26.83_{\pm2.26}$ | $36.03_{\pm1.65}$ | $53.07_{\pm1.15}$ | $60.73_{\pm0.77}$ | $70.22_{\pm0.00}$ |
| CosLayer | Core10Lifelong | $31.45_{\pm10.30}$ | $43.25_{\pm9.15}$ | $52.17_{\pm10.29}$ | $65.40_{\pm5.68}$ | $78.76_{\pm0.36}$ |
| WeightNorm | Core10Lifelong | $68.52_{\pm4.46}$ | $76.78_{\pm1.44}$ | $81.38_{\pm0.89}$ | $83.89_{\pm0.62}$ | $86.29_{\pm0.29}$ |
| OriginalWeightNorm | Core10Lifelong | $67.50_{\pm5.30}$ | $75.86_{\pm1.15}$ | $80.33_{\pm0.89}$ | $82.42_{\pm0.75}$ | $86.07_{\pm0.51}$ |
| CosLayer-Masked | Core10Lifelong | $65.58_{\pm4.18}$ | $72.28_{\pm2.19}$ | $76.89_{\pm1.12}$ | $78.69_{\pm0.66}$ | $79.56_{\pm0.65}$ |
| WeightNorm-Masked | Core10Lifelong | $42.05_{\pm5.58}$ | $48.26_{\pm4.63}$ | $54.12_{\pm6.72}$ | $57.78_{\pm4.81}$ | $58.74_{\pm16.05}$ |
| OriginalWeightNorm-Masked | Core10Lifelong | $26.76_{\pm5.72}$ | $34.75_{\pm5.45}$ | $30.12_{\pm11.94}$ | $31.73_{\pm7.00}$ | $46.05_{\pm4.96}$ |
| Linear | Core10Lifelong | $68.47_{\pm4.57}$ | $76.33_{\pm1.32}$ | $80.79_{\pm0.91}$ | $83.08_{\pm0.57}$ | $86.19_{\pm0.33}$ |
| Linear-no-bias | Core10Lifelong | $68.47_{\pm4.57}$ | $76.33_{\pm1.32}$ | $80.79_{\pm0.91}$ | $83.08_{\pm0.57}$ | $86.19_{\pm0.33}$ |
| Linear-Masked | Core10Lifelong | $23.17_{\pm6.06}$ | $24.29_{\pm6.54}$ | $29.25_{\pm7.45}$ | $30.79_{\pm10.56}$ | $38.08_{\pm28.04}$ |
| Linear-no-bias-Masked | Core10Lifelong | $23.11_{\pm5.98}$ | $24.06_{\pm6.49}$ | $28.82_{\pm7.34}$ | $30.76_{\pm10.48}$ | $38.27_{\pm28.03}$ |
| KNN | Core10Lifelong | $55.05_{\pm4.00}$ | $64.80_{\pm2.23}$ | $72.20_{\pm2.05}$ | $76.51_{\pm1.55}$ | $75.71_{\pm1.50}$ |
| SLDA | Core10Lifelong | $63.00_{\pm3.03}$ | $63.96_{\pm2.35}$ | $35.55_{\pm1.38}$ | $73.46_{\pm0.82}$ | $88.06_{\pm0.01}$ |
| MeanLayer | Core10Lifelong | $65.06_{\pm4.08}$ | $71.30_{\pm2.70}$ | $76.25_{\pm1.37}$ | $78.33_{\pm0.49}$ | $79.61_{\pm0.00}$ |
| MedianLayer | Core10Lifelong | $61.46_{\pm4.26}$ | $69.28_{\pm2.51}$ | $74.81_{\pm1.24}$ | $76.69_{\pm0.85}$ | $78.16_{\pm0.03}$ |
| CosLayer | CUB200 | $5.19_{\pm0.77}$ | $5.41_{\pm1.01}$ | $6.77_{\pm1.19}$ | $8.74_{\pm1.94}$ | $30.00_{\pm0.73}$ |
| WeightNorm | CUB200 | $11.13_{\pm1.19}$ | $15.78_{\pm0.83}$ | $25.39_{\pm0.98}$ | $34.96_{\pm0.93}$ | $54.41_{\pm0.50}$ |
| OriginalWeightNorm | CUB200 | $11.17_{\pm0.87}$ | $16.16_{\pm0.96}$ | $26.10_{\pm0.93}$ | $35.50_{\pm0.57}$ | $52.90_{\pm0.67}$ |
| CosLayer-Masked | CUB200 | $11.43_{\pm1.20}$ | $16.85_{\pm0.41}$ | $26.79_{\pm0.77}$ | $36.03_{\pm0.74}$ | $33.07_{\pm0.63}$ |
| WeightNorm-Masked | CUB200 | $7.51_{\pm1.02}$ | $9.10_{\pm1.04}$ | $12.37_{\pm1.37}$ | $14.40_{\pm0.92}$ | $16.24_{\pm0.85}$ |
| OriginalWeightNorm-Masked | CUB200 | $3.96_{\pm1.02}$ | $3.92_{\pm0.82}$ | $4.39_{\pm0.74}$ | $4.96_{\pm1.26}$ | $5.75_{\pm1.02}$ |
| Linear | CUB200 | $10.88_{\pm0.82}$ | $15.38_{\pm0.86}$ | $24.27_{\pm0.86}$ | $33.29_{\pm0.93}$ | $28.84_{\pm0.75}$ |
| Linear-no-bias | CUB200 | $10.73_{\pm0.82}$ | $15.07_{\pm0.73}$ | $24.23_{\pm0.86}$ | $33.49_{\pm0.91}$ | $28.92_{\pm0.78}$ |
| Linear-Masked | CUB200 | $2.07_{\pm0.53}$ | $2.53_{\pm0.57}$ | $2.55_{\pm0.55}$ | $2.77_{\pm0.42}$ | $2.24_{\pm0.24}$ |
| Linear-no-bias-Masked | CUB200 | $2.42_{\pm0.33}$ | $2.15_{\pm0.43}$ | $2.36_{\pm0.59}$ | $2.72_{\pm0.58}$ | $2.08_{\pm0.29}$ |
| KNN | CUB200 | $10.33_{\pm1.22}$ | $14.31_{\pm0.56}$ | $20.41_{\pm1.07}$ | $25.54_{\pm0.75}$ | $43.79_{\pm0.00}$ |
| SLDA | CUB200 | $1.94_{\pm0.65}$ | $2.37_{\pm0.70}$ | $3.73_{\pm0.44}$ | $15.93_{\pm2.68}$ | $58.61_{\pm0.12}$ |
| MeanLayer | CUB200 | $10.37_{\pm1.25}$ | $15.07_{\pm0.46}$ | $25.24_{\pm1.05}$ | $34.55_{\pm0.74}$ | $54.14_{\pm0.00}$ |
| MedianLayer | CUB200 | $10.77_{\pm1.02}$ | $14.92_{\pm0.54}$ | $23.07_{\pm0.90}$ | $31.91_{\pm1.00}$ | $53.28_{\pm0.00}$ |

## F   VISUALIZATIONS

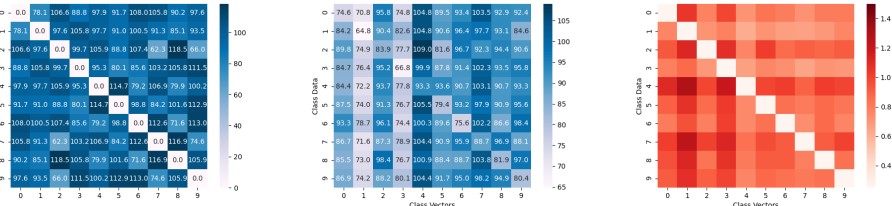

(a) Linear: Angles between the different vectors of the output layer

(b) Linear: Mean Angles between the vectors and data class by class.

(c) Linear: Interference Risks



(d) WeightNorm: Angles between the different vectors of the output layer

(e) WeightNorm: Mean Angles between the vectors and data class by class.

(f) WeightNorm: Interference Risks

## G   TASK ORDERS

```
from continuum.scenarios import create_subscenario
import numpy as np

# we suppose scenario, num_tasks and seed already set
np.random.seed(seed)
task_order = np.arange(num_tasks)
scenario = create_subscenario(scenario, task_order)
```

(a) Code to change task order of the original scenario. We used seeds $[0, 1, 2, 3, 4, 5, 6, 7]$. Note: for seed 0, we use original order without modification.

## H    SCENARIO REPRODUCIBILTY

```python
from continuum.datasets import Core50
from continuum import ClassIncremental

dataset = dataset = Core50("./datasets",
                           download=True,
                           train=train)
scenario = ClassIncremental(dataset,
                            nb_tasks=10,
                            transformations=transform)
```

(a) Code for Core50 scenario

```python
from continuum.datasets import Core50
from continuum import ContinualScenario

dataset = Core50("./datasets",
                 scenario="domains",
                 classification="category",
                 train=train)
scenario = ContinualScenario(dataset,
                             transformations=transform)
```

(b) Code for Core10Lifelong scenario

```python
from continuum.datasets import Core50
from continuum import ContinualScenario

dataset = Core50("./datasets",
                 scenario="objects",
                 classification="category",
                 train=train)
scenario = ContinualScenario(dataset,
                             transformations=transform)
```

(c) Code for Core10Mix scenario

```python
from continuum.datasets import CUB200
from continuum import ClassIncremental

dataset = CUB200("./datasets",
                 train=train)
scenario = ClassIncremental(dataset,
                            nb_tasks=10,
                            transformations=transform)
```

(d) Code for CUB200 scenario

```python
from continuum.datasets import CIFAR100
from continuum import ContinualScenario

dataset = CIFAR100("./datasets",
                   labels_type="category",
                   task_labels="lifelong",
                   train=train)
scenario = ContinualScenario(dataset,
                             transformations=transform)
```

(e) Code for CIFAR100Lifelong scenario

Figure 8: Code to reproduce the scenarios used in the paper with continuum library.

