# OpenReview forum: "Continual Learning in Deep Networks: an Analysis of the Last Layer"
_ICLR.cc/2022/Conference — ICLR 2022 Submitted_

### Official Review · Reviewer_e1ZF · 2021-10-27

**Correctness:** 3
**Technical Novelty And Significance:** 2
**Empirical Novelty And Significance:** 2
**Recommendation:** 3
**Confidence:** 4

**Main Review:**

1. I am not clear on why using neural networks became essential in this study.

Given that the feature extractor part of a given neural network is frozen, and the paper doesn't provide any insight on feature learning during continual learning, I'd recommend the authors to start from simple linear models, including logistic regression, and SVMs. In that case, the impact of the neural network architectures can be ruled out, and the conclusion could be clearer.

2. As mentioned later in the paper, arguments made regarding the lengths and the biases of vectors in the weight matrix in the last layer don't seem consistent across various settings and datasets in continual learning.

Personally, I think the main issue is that the learnt feature spaces are drastically different within the selected models, and the confidence of the top linear layer in classification also varies, therefore, when only studying the top linear layers, behaviours may not be consistent.

The secondary issue is that the feature vectors are all in very high-dimensional space where, with high probability, vectors are equal-distant from each other, which causes trouble and issues when analysing their relations. That probably also explains why the proposed 'MedianLayer' performed similarly to the previously proposed 'MeanLayer'

3. From the experiments and analyses in the paper, as also is mentioned in the discussion and conclusion section, we can see that classic prototype-based approaches still stand out, and the proposed methods don't add much more value to them, either performance or understanding in terms of their behaviours in continual learning.

**Summary Of The Paper:**

the paper analysed behaviours of the last linear layer of pre-trained convolutional neural networks in the settings of continual learning where either new instances are provided in new episodes or new classes are given in new episodes. Based on arguments and observations provided in the paper, the authors proposed to constrain the weight matrix in the last linear layer to only have unit-length vectors, so that the lengths and biases of these vectors won't introduce any semantic drifting in continual learning.

**Summary Of The Review:**

I have major concerns about the clarity of messages conveyed in the paper, and also the insights that this paper could potentially provide on continual learning if the paper was accepted. Therefore, I do not recommend accepting this submission.

---

> ### Author Response · Authors · 2021-11-17
> **Answer Reviewer BTLN**
>
> >I'd recommend the authors to start from simple linear models, including logistic regression, and SVMs. In that case, the impact of the neural network architectures can be ruled out, and the conclusion could be clearer.
>
> It could an interesting baseline indeed, thanks for the idea. We however thought that it would be very relevant to study linear models using the actual feature space obtained by training a NN. Our point is precisely not to rule out the NN architecture, since linear models can perform very differently depending on the (e.g.) the covariance of the examples in the feature space.
>
> >The secondary issue is that the feature vectors are all in very high-dimensional space where, with high probability, vectors are equal-distant from each other, which causes trouble and issues when analyzing their relations. That probably also explains why the proposed 'MedianLayer' performed similarly to the previously proposed 'MeanLayer'
>
> What do you mean by equal-distant from each others? The latent space is not very problematic in our setting since we can reach a good iid setting.
>
> >As mentioned later in the paper, arguments made regarding the lengths and the biases of vectors in the weight matrix in the last layer don't seem consistent across various settings and datasets in continual learning. Personally, I think the main issue is that the learnt feature spaces are drastically different within the selected models, and the confidence of the top linear layer in classification also varies, therefore, when only studying the top linear layers, behaviors may not be consistent.
>
> We are looking for consistency into data distribution drifts in this paper (and we have consistency in our results wrt data distribution drifts), however indeed the distribution of features is always determinant for a good performing classifier...
>
> >[...] we can see that classic prototype-based approaches still stand out, and the proposed methods don't add much more value to them, either performance or understanding in terms of their behaviours in continual learning.
>
> This is true, however, prototype-based method are not well suited when models are trained end-2-end. We acknowledge their performance in our setting, but we believe that using other methods could be more suited for end-2-end training.

---

### Official Review · Reviewer_BTLN · 2021-11-01

**Correctness:** 3
**Technical Novelty And Significance:** 1
**Empirical Novelty And Significance:** 2
**Recommendation:** 5
**Confidence:** 4

**Main Review:**

Clarity:

Overall writing quality of the paper can be improved a lot. Especially the Sec5 Results which needs to highlight the current
contributions and can explained bit more in-detailed. The method proposed is well described and it would take some time to have reproduce the results.

Novelty:

The methodological contributions do not seem to be novel and all the results are as expected w.r.t  the current
line of research of doing continual learning keeping the feature extractor part of the network fixed.

Pros:
Very well detailed experimental setup covering all the scenarios. Authors evaluated the proposed ideas on Incremental Scenarios(CIFAR10,Core50,CUB200), Lifelong Learning Scenarios(Core10LifeLong, CIFAR100LifeLong) and Mixed Scenario(Cire10Mix)

Cons:

As mentioned the Novelty is pretty much limited and results as as expected w.r.t current line of research.



**Summary Of The Paper:**

The authors analyzed the effect of changes to the last output
layers of the neural network in the continual, incremental learning scenario
and effects of it two scenarios 1. weight modification 2.Inference.
Authors also claim to propose simplified weight normalization layer, two masking strategies, and an alternative to NearestMean Classifier using median vectors to address catastrophic forgetting in the output layer in both continual, Incremental scenarios.

During the experiment setup authors separated the network into two parts

1. feature extractor part consisting of all but the final layer (kept fixed to avoid projection drift)
2. classifier part which was given by the output layer.[Much of the analysis is concentrated on Part-B]

**Summary Of The Review:**

Please refer before sections

Changes:
In the Sec 5.2 (Median Layer...): Third line "MeanLayer Could Show some more advantages against MeanLayer"

---

> ### Author Response · Authors · 2021-11-17
> **Answer to Reviewer BTLN**
>
> >The authors analyzed the effect of changes to the last output layers of the neural network in the continual, incremental learning scenario and effects of it two scenarios 1. weight modification 2.Inference
>
> Weight modification and interference are not scenarios they are the causes of catastrophic forgetting. We study them into three scenarios we called *incremental*, *lifelong* and *mixed*.
>
> >The methodological contributions do not seem to be novel
>
> Could you provide a continual learning paper that study how learning behave depending on how the data distribution drift happens?
>
> >the results are as expected w.r.t the current line of research of doing continual learning keeping the feature extractor part of the network fixed.
>
> We respectfully disagree with the reviewer: Training a linear layer (weightnorm) with SGD in an incremental learning without encountering catastrophic forgetting is not an already known result, even keeping the feature extractor part of the network fixed.
>
>
> > In the Sec 5.2 (Median Layer...): [...]
>
> Thanks for pointing out the typo section 5.2.

---

### Official Review · Reviewer_Kc24 · 2021-11-02

**Correctness:** 3
**Technical Novelty And Significance:** 2
**Empirical Novelty And Significance:** 2
**Recommendation:** 3
**Confidence:** 4

**Main Review:**

The paper isolates the output layer for study yet in continual learning all layers are updated. No concrete motivation is provided as to why the output layer should be studied in isolation.

Moreover, some of the observations highlighted in the paper may not hold if the neural network is allowed to be updated and the neural network is studied holistically.

Furthermore, most observation (e.g., _masking is efficient in incremental settings to avoid weight modifications_) are well known from prior work. Most observations are not contrasted with prior work.

**Summary Of The Paper:**

This paper studies the output layer in a deep neural network in the continual learning setting. The paper studies multiple types of output layers under different continual learning configurations. Some suggestions are made based on observed empirical trends.

**Summary Of The Review:**

The validity and novelty of the results and observations in the paper are somewhat dubious.

---

> ### Author Response · Authors · 2021-11-17
> **Answer Reviewer Kc24**
>
> >No concrete motivation is provided as to why the output layer should be studied in isolation.
>
> Sorry if we did not make clear why the classifier is important for supervised continual learning.
> The output layer is very important for the prediction: it converts the representation into a prediction. In continual learning litterature, many papers found out that catastrophic forgetting in the output layer is determinant (cf section 2 for references). A better understanding of the layer in particular could lead to improvement in continual learning approaches. The isolation enables a more precise analysis. Moreover, the problem of continual learning can be decomposed into learning representation and learning a classifier without any supplementary constraints.
>
> > some of the observations highlighted in the paper may not hold if the neural network is allowed to be updated
>
> This is true, this is why we work under the assumption that the feature exctractor is frozen for the analysis. The full training of the model is more complicated but it might be helpful to understand the different components more precisely to improve the training of the complete model.
>
> >Furthermore, most observation (e.g., masking is efficient in incremental settings to avoid weight modifications) are well known from prior work. Most observations are not contrasted with prior work
>
> Could you kindly provide references to those prior works?

---

### Official Review · Reviewer_uN9P · 2021-11-02

**Correctness:** 3
**Technical Novelty And Significance:** 2
**Empirical Novelty And Significance:** 2
**Recommendation:** 3
**Confidence:** 4

**Main Review:**

Strength:
+ The summary of different design choices, including the weight normalization,
+ Some illustrations are valuable for understanding the cause of forgetting, eg. the norm, bias, and their changes during the incremental process; and the interference plots.

Weakness:
-  The assumption on non-projection-drift is not very realistic. In continual learning problems, it is very often that the feature representations are shifted over time, along with new data streams.  However, under the paper's assumption, it requires a fixed feature extractor.
-  Since the study is based on a non-projection-drift setting, it requires pre-trained models as the feature extraction as stated in the experiment part, which is not often practical in many settings. Although it does not use the pre-trained models trained on the target dataset, it is still unfair to compare with other works in the literature.
- For Figure 2 (interference), what is the root cause for the interference? Why the mean angle of z and A_i (middle) is not as discriminative as the angles among A_i (left).
- The paper proposed a modified version of the weight normalization layer, but the proposed modified layer operation does not show clear improvement but slight drop over the original designs in most cases (weight_norm vs original_weight_norm in Table 1). Similar to the proposed median layer vs original mean layer.
-  The proposed masking approach is not stable and tends to produce bad performance.
- The figure plots of accuracy are a bit cluttered and hard to read.

**Summary Of The Paper:**

This paper presents a study on different last output layers, under the continual learning problem. More specifically, the study is conducted in a controlled environment without projection shift. The paper proposed several design choices of the last layer, including a modified weight normalization layer, masking operations, and median vector. Different scenarios of incremental/lifelong and mixed learning have been experimented with.

**Summary Of The Review:**

The study of the feature drift is important, otherwise, the study has to rely on the pre-trained feature model, which makes the setting impractical in many cases.
The paper provides some summarization of multiple tracks to deal with the weights normalization and preserving in the last layer but the insights brought from the work are limited. The analysis is strait-forward on the observation of weights but there is not much deeper and richer analysis on the causes of such observation.
The experiment results do not support most hypotheses made, but the cause of the failures remains unclear.

---

> ### Author Response · Authors · 2021-11-17
> **Answer Reviewer uN9P**
>
>
> >The assumption on non-projection-drift is not very realistic. In continual learning problems, it is very often that the feature representations are shifted over time, along with new data streams. However, under the paper's assumption, it requires a fixed feature extractor.
>
> The paper is not about an approach for continual learning it is about the analysis of the last layer, the setting is not build to be realistic but to make possible such analysis.
>
> >Since the study is based on a non-projection-drift setting, it requires pre-trained models as the feature extraction as stated in the experiment part, which is not often practical in many settings. Although it does not use the pre-trained models trained on the target dataset, it is still unfair to compare with other works in the literature.
>
> We do not compare with other work in other data settings (we agree it would be unfair). We only compare with results of our own experiments settings and all of them have been produced under the same conditions.
>
> >For Figure 2 (interference), what is the root cause for the interference? Why the mean angle of z and A_i (middle) is not as discriminative as the angles among A_i (left).
>
> The root cause of interference is a lower angle (Z_{c=i}, A_{j!=i}) than angle (Z_{c=i}, A_{i}). The values in the figure of the right depend on both angles while values of the central plot depend only on the angle (Z_{c=i}, A_{j}).
>
> >The paper proposed a modified version of the weight normalization layer, but the proposed modified layer operation does not show clear improvement but slight drop over the original designs in most cases (weight_norm vs original_weight_norm in Table 1). Similar to the proposed median layer vs original mean layer.
>
> Table 1 is only a subset of our experiments, in incremental experiments and mixed experiment (Fig. 3 a,c,e,f) the proposed weightnorm largely improves the original weightnorm. Indeed, median layer does not improves much mean layer, this is somehow a negative result.
>
> > The proposed masking approach is not stable and tends to produce bad performance.
>
> This is true, however, in incremental scenario masking can be efficient. It might be legitimate to avoid masking because of some instability even if it is efficient in some settings. Nevertheless, this experiment is valuable since it shows that depending on the data distribution drifts an approach can be efficient in one case and not in the other. This type of result is rare in the continual learning literature.
>
> *The study of the feature drift is important*
> We completely agree, the goal of this paper is not to claim that feature drift should not be addressed, it is just not the topic of the paper. Approach that address feature drift are complementary to our work.
>
> > The experiment results do not support most hypotheses made
>
> We will appreciate from you to cite some example of claims not supported so we can address them.

---

### Author Response · Authors · 2021-11-17
**General Answer**

Dear reviewers, thank you for your review. You point out various concerns, we tried to answer most of them them point by point in personal answer. We hope that it will make more clear what the paper is about to have a fruitful discussion. Even if obviously it should not lead to an acceptance here, we hope it will help us to improve the paper.

---

### Decision · Program_Chairs · 2022-01-20

**Decision:**

Reject

**Comment:**

The paper proposes an empirical study on the effect of various types of output layers of deep neural networks in different scenarios of continual learning. The authors draw several insights, such as ways of selecting the best output layer depending on type of scenario and a description of the different sources of performance drop (forgetting, interference, and projection drifts). The paper proposes different ways of mitigating catastrophic forgetting: a weight normalization layer, two masking strategies, and a variant of NMC using median vectors.

The paper presented a detailed experimental setup covering a large number of scenarios of continual learning: incremental, Lifelong Learning and Mixed Scenario. This was highlighted by Reviewer BTLN, and the AC agrees.

The main point of criticism for the work is the lack of novelty and the low significance of the findings. These were highlighted by all four reviewers.

Perhaps the aspect limiting significance is the fact that the feature extractors are assumed to be fixed, which is unlike most interesting settings in continual learning. This was mentioned by reviewers BTLN, uN9P, e1ZF. It is unclear whether the findings provided in this work would generalize to that setting. On that note, Reviewer e1ZF points out that not adapting the feature extractor could be the source of some inconsistencies observed. Studying this further would improve the work.

Overall, all four reviewers recommend rejecting the paper. The AC agrees with this decision and encourages the authors to consider extending the analysis to situations where the feature extractor is not fixed.